# Tumor-Intrinsic Enhancer of Zeste Homolog 2 Controls Immune Cell Infiltration, Tumor Growth, and Lung Metastasis in a Triple-Negative Breast Cancer Model

**DOI:** 10.3390/ijms25105392

**Published:** 2024-05-15

**Authors:** Lenore Monterroza, Maria M. Parrilla, Sarah G. Samaranayake, Dormarie E. Rivera-Rodriguez, Sung Bo Yoon, Ramireddy Bommireddy, Justin Hosten, Luisa Cervantes Barragan, Adam Marcus, Brian S. Dobosh, Periasamy Selvaraj, Rabindra Tirouvanziam

**Affiliations:** 1Department of Pediatrics, Emory University School of Medicine, Atlanta, GA 30322, USA; lenore.monterroza@emory.edu (L.M.); maria.milagros.parrilla@emory.edu (M.M.P.); justin.hosten@emory.edu (J.H.); brian.seth.dobosh@emory.edu (B.S.D.); 2Department of Pathology & Laboratory Medicine, Emory University School of Medicine, Atlanta, GA 30322, USA; ramireddy.bommireddy@emory.edu; 3MD Program, Emory University School of Medicine, Atlanta, GA 30322, USA; sarah.samaranayake@emory.edu; 4Department of Microbiology & Immunology, Emory University School of Medicine, Atlanta, GA 30322, USA; dormarie.rivera@emory.edu (D.E.R.-R.); lcervantes@emory.edu (L.C.B.); 5Department of Hematology & Medical Oncology, Emory University School of Medicine, Atlanta, GA 30322, USA; sung.bo.yoon@emory.edu (S.B.Y.); aimarcu@emory.edu (A.M.)

**Keywords:** CD4+ T cell, CD8+ T cell, invasion, myeloid-derived suppressor cell, triple-negative breast cancer, tumor-infiltrating neutrophils

## Abstract

Triple-negative breast cancer (TNBC) is an aggressive and highly metastatic type of tumor. TNBC is often enriched in tumor-infiltrating neutrophils (TINs), which support cancer growth in part by counteracting tumor-infiltrating lymphocytes (TILs). Prior studies identified the enhancer of zeste homolog 2 (EZH2) as a pro-tumor methyltransferase in primary and metastatic TNBCs. We hypothesized that EZH2 inhibition in TNBC cells per se would exert antitumor activity by altering the tumor immune microenvironment. To test this hypothesis, we used CRISPR to generate *EZH2* gene knockout (KO) and overexpressing (OE) lines from parent (wild-type—WT) 4T1 cells, an established murine TNBC model, resulting in EZH2 protein KO and OE, respectively. In vitro, EZH2 KO and OE cells showed early, transient changes in replicative capacity and invasiveness, and marked changes in surface marker profile and cytokine/chemokine secretion compared to WT cells. In vivo, EZH2 KO cells showed significantly reduced primary tumor growth and a 10-fold decrease in lung metastasis compared to WT cells, while EZH2 OE cells were unchanged. Compared to WT tumors, TIN:TIL ratios were greatly reduced in EZH2 KO tumors but unchanged in EZH2 OE tumors. Thus, EZH2 is key to 4T1 aggressiveness as its tumor-intrinsic knockout alters their in vitro secretome and in vivo primary tumor growth, TIN/TIL poise, and metastasis.

## 1. Introduction

Breast cancer is a global health problem affecting 2.3 million individuals globally [1]. Defined as estrogen- and progesterone-receptor negative and lacking HER2 overexpression, triple-negative breast cancer (TNBC) holds the poorest prognosis among breast cancer types [2]. This aggressive form occurs in 15–20% of patients, accounting for ~170,000 cases worldwide [3]. TNBC is comprised of different subtypes, characterized by distinct molecular signatures. Common treatments include chemotherapy, radiation, and surgical resection. Patients with T-cell-rich or non-refractory (“hot”) tumors also benefit from newly developed immune checkpoint inhibitor (ICI) and chimeric antigen receptor (CAR) T-cell therapies [4]. Unfortunately, a large proportion of TNBC patients are refractory to ICI therapy (designated as “cold”) [3,5]. Consequently, primary tumors and metastatic outgrowth from chemotherapy-resistant TNBC are a major cause of mortality [6].

Genetic and epigenetic alterations in TNBC are some of the main obstacles for successful responses to therapy [7]. Among the TNBC markers identified as potential therapeutic targets, the methyltransferase enhancer of zeste homolog 2 (EZH2) holds significant promise, as its overexpression is associated with poor prognosis and short disease-free survival in patients [8,9,10,11]. This may occur in part through the increased stability of EZH2 due to upstream regulatory pathways leading to its post-translational modification in aggressive TNBCs [12]. Upregulated EZH2 contributes to tumor development, progression, and metastasis via multiple downstream pathways, including but not limited to the modulation of stimulator of interferon genes (STINGs) [13], transforming growth factor β (TGF β) [14], signal transducer and activator of transcription 3 (STAT3) [15], and Wnt [16] signaling. Thus, EZH2 inhibitors have been tested in combination with ICIs and other chemotherapies [17]. A confounding factor, however, is that the expression of EZH2 occurs in both tumor cells and in tumor-associated innate and adaptive immune cells (e.g., tumor-infiltrating neutrophils (TINs) and lymphocytes (TILs), respectively) [7].

In this study, we probed the role of EZH2 in primary and metastatic TNBCs [18] using a 4T1 murine TNBC model. EZH2 was previously shown to be significantly upregulated in 4T1 compared to normal mouse breast epithelial cells [10]. To this end, we used CRISPR technology to drive *EZH2* gene knockout (KO) and overexpression (OE) in stable cell lines derived from parent wild-type (WT) 4T1 cells, resulting in EZH2 protein KO and OE, respectively. While the replicative and invasive capacities of EZH2 KO and OE cells did not broadly differ from those of WT 4T1 cells in vitro, the former but not the latter showed significantly decreased primary growth and lung metastasis in vivo, along with dramatic reductions in the ratios of TINs to both CD4+ and CD8+ T cells.

## 2. Results

### 2.1. EZH2 Knockout and Overexpressing Lines Derived from the 4T1 TNBC Model Behave Similarly to the Parent Line In Vitro

We generated multiple *EZH2* gene KO and *EZH2* gene OE clones from parent WT 4T1 cells using CRISPR/Cas9 and gene amplification followed by a round of sorting. Lysates were prepared from each clone and analyzed by Western blot, using α-tubulin as a normalization control. All KO and OE clones demonstrated a successful elimination and overexpression of the EZH2 protein, respectively (Figure 1).

To characterize the effect of altered cell-intrinsic EZH2 expression on 4T1 behavior in vitro, we selected 4T1 EZH2 KO11 and EZH2 OE6 clones for further investigation. These two clones were selected because of their grossly normal morphology and viability compared to the WT. They were plated in parallel with the parent WT 4T1 cell line at 5 × 10^5^ cells, and counts were performed at 24, 48, and 72 h post-plating, revealing no significant difference in replication (Figure 2A). Next, we used a 3D spheroid invasion assay to evaluate the invasive capacities of EZH2 KO, EZH2 OE, and WT 4T1 cells at 0, 24, 48, and 72 h post-plating (Figure 2B). EZH2 OE cells showed increased spheroid circularity at 24, 48, and 72 h (Figure 2C), but a decreased invasive area (another measure of invasiveness) at 24 h (Figure 2D). Conversely, compared to the WT cells in vitro, the EZH2 KO cells showed decreased spheroid circularity at 24 h (Figure 2C), but increased invasive areas at 0 and 24 h (Figure 2D). Note however that at 72 h, WT, EZH2 OE, and EZH2 KO cells showed equal invasive areas (Figure 2D). 

### 2.2. EZH2 Expression Impacts Surface Phenotype and Secreted Mediators of 4T1 Cells In Vitro

Using flow cytometry (Appendix A), the expression of relevant markers on EZH2 KO, EZH2 OE, and WT 4T1 cells was assessed against unstained controls for positivity (Appendix A), and positively expressed markers were then quantified over multiple repeats across lines (Figure 3A). The expression levels of CD24, CD44, ICAM-1, MHC-I, and checkpoint inhibitor PD-L1 differed across 4T1 WT, EZH2 KO, and EZH2 OE cells. Specifically, expressions of CD24 and CD44 were lower on EZH2 KO compared to both the WT and EZH2 OE cells, and expressions of ICAM-1 and PD-L1 were lower in EZH2 KO than in EZH2 OE cells, while EZH2 OE cells were similar to the WT for all markers except for a lower MHC-I expression. When assessing their profiles of twelve secreted mediators (Figure 3B), we observed no significant difference between the WT or either EZH2 KO or EZH2 OE cells. However, EZH2 KO cells secreted higher levels of several mediators in comparison to EZH2 OE cells, with significant >4-fold increases for GM-CSF and MCP-1 and a >2-fold increase for IL1β. Similar trends for the increased secretion in EZH2 KO vs. EZH2 OE cultures were observed for IFNα, IL-10, IP-10, and TNFα secretion, albeit non-significant.

### 2.3. EZH2 Knockout Reduces Primary Tumor Growth and Lung Metastasis of 4T1 Cells In Vivo

Because the in vivo growth of tumor cells occurs at longer intervals and also involves other factors (e.g., immune cells), the in vitro data on EZH2 OE and KO 4T1 cells may not be predictive of their growth and metastatic potential in animals. Thus, we next moved to test the impacts of EZH2 knockout and overexpression on 4T1 TNBC primary tumor growth and metastasis in vivo. To this end, we challenged mice with WT, EZH2 KO, and EZH2 OE cells. The cells were injected subcutaneously in the flank of BALB/c mice and allowed to grow over 21 days (Figure 4A). While the WT and EZH2 OE cells showed similar primary tumor growth across all timepoints, EZH2 KO cells grew significantly slower than both, resulting in a 2-fold-smaller size at day 21 (Figure 4B). Since the 4T1 TNBC model is spontaneously metastatic, the lungs of tumor-bearing mice were isolated at the end of the in vivo challenge and processed for the quantification of metastatic cells. Strikingly, EZH2 KO showed a significantly reduced (approximately 10-fold lower) lung metastatic burden compared to the WT and EZH2 OE cells (Figure 4C). To confirm these results, we repeated the in vivo challenge comparing WT and EZH2 KO cells and extended the duration by a week. Again, EZH2 KO cells grew significantly slower than the WT, which was evident macroscopically by day 18 (Appendix A). The difference in primary tumor size between EZH2 KO and WT groups reached 4-fold by day 28 (Appendix A). Additionally, a 10-fold lower lung metastatic burden was again observed between EZH2 KO and WT cells (Appendix A). To probe the potential implication of the EZH2-regulated STING pathway in the observed effects, we conducted parallel in vivo challenge with the longitudinal treatment of EZH2 KO- and WT-cell-injected animals with the STING agonist MSA-2. We observed that the MSA-2 agonism of STING significantly decreased the primary tumor growth (Appendix A) and metastatic potential (Appendix A) of WT cells. Interestingly, MSA-2 treatment did not decrease the primary tumor growth and metastatic potential in EZH2 KO cells compared to the WT.

### 2.4. Tumor-Intrinsic EZH2 Knockout Alters the Balance of Neutrophils and CD4+ and CD8+ T Cells in Primary 4T1 Tumors

To determine whether altering EZH2 expression affects immune cell infiltration into tumors, we prepared single-cell suspensions from WT, EZH2 KO, and EZH2 OE primary tumors, and quantified live infiltrating leukocytes through flow cytometry analysis and the sequential gating of relevant subsets (Figure 5A). While WT and EZH2 OE tumors had similar proportions across all leukocyte subsets measured, EZH2 KO tumors showed on average a 2-fold-lower proportion of neutrophils, and 2- and 10-fold-higher proportions of CD4+ and CD8+ T cells, respectively, than the WT and EZH2 OE tumors (Figure 5B). Consistently, EZH2 KO tumors showed on average 3- to 4-fold-higher proportions of CD3+ T cells (including both CD4+ and CD8+), while their lower proportions of infiltrated neutrophils included both mature and immature cells (Appendix A). The paradoxical effect of tumor-intrinsic EZH2 knockout on infiltrated neutrophils (decrease) as well as CD4+ and CD8+ T cells (increase) was even more striking when expressed as ratios, revealing a >20-fold decrease in the neutrophil:CD8+ T-cell ratio in EZH2 KO vs. WT and EZH2 OE tumors (Figure 5B).

## 3. Discussion

In this study, we report the successful CRISPR-aided generation of several EZH2 KO and EZH2 OE clones derived from the 4T1 murine TNBC cell line. Based on grossly normal morphology and viability compared to the WT, one EZH2 KO line and EZH2 OE line were selected for further phenotypic evaluations in vitro and in vivo. An analysis of in vitro proliferation on 2D plates and invasiveness in a 3D spheroid assay [19] showed no major differences between EZH2 KO, EZH2 OE, and WT 4T1 cell lines, suggesting the little regulatory impact of EZH2 on these properties. These results are consistent with prior data on the siRNA-aided knockdown of EZH2 in 4T1 cells, also reporting no apparent effect on cell proliferation or invasiveness [14]. Our study is the first to introduce EZH2 OE cells and shows that these and WT cells behaved very similarly, both in vitro and in vivo, suggesting that EZH2 expression may be already saturating with regards to downstream signaling in parent WT 4T1 cells.

In regard to the cell surface markers, only MHC class I was altered (lowered) in EZH2 OE cells compared to the WT, while EZH2 KO cells showed a decreased expression of multiple immune activation markers (CD24, CD44, ICAM, and PD-L1) compared to EZH2 OE cells, with intermediate levels in WT cells. The significantly altered secretion of GM-CSF, MCP-1, and IL1β (all myeloid mediators) was observed in EZH2 KO culture supernatants, suggesting that EZH2 may impact immune crosstalk by 4T1 cells. Indeed, EZH2 KO tumors grown in vivo decreased the proportion of infiltrated neutrophils and increased those of infiltrated CD4+ and CD8+ T cells (culminating in a 20-fold reduction in the neutrophil:CD8+ T-cell ratio). Concomitantly, EZH2 KO tumors displayed significantly reduced growth in the primary tumor site (by 2 to 4 fold) and lung metastatic potential (by 10 fold). Together, our findings suggest that, while baseline EZH2 expression in 4T1 TNBC cells does not seem to play a critical role in vitro, it is necessary to maintain high TIN:TIL ratios and their growth and metastatic potential in vivo.

There is ample evidence that EZH2 is involved in the epigenetic control of critical immune regulatory pathways [7], notably including the STING pattern recognition receptor [13,20]. STING plays multiple crucial roles in danger signaling, interferon secretion, and leukocyte infiltration in solid tumors [21]. Prior research on the 4T1 model demonstrated altered levels of pro-inflammatory and interferon-related cytokines in serum [22,23]. Our in vivo experiments using the STING agonist MSA-2 as a longitudinal treatment combined with either 4T1 WT or EZH2 KO cells showed significant reductions in primary tumor growth and metastasis potential for the former, but not the latter. Together, these findings suggest that high EZH2 expression in WT 4T1 cells may act in part via STING inhibition and can be overcome by MSA-2 treatment, while STING activity may be fully released in EZH2 KO cells, explaining the absence of additive antitumor effects of MSA-2. Future studies on our EZH2 KO and EZH2 OE lines will evaluate the relative roles of STING and other regulators of immune signaling by 4T1 cells.

Our in vivo findings are consistent with previous studies demonstrating that cells with high EZH2 expressions have an advantage in metastasizing, while EZH2 KO 4T1 tumor-bearing mice have significantly longer survival and decreased occurrence of metastatic colonies [24]. Previous work showed that metastasis in the 4T1 model is impacted by the capacity of circulating cancer cells to undergo epithelial–mesenchymal transition and successfully establish micrometastasis at a distant organ site [25]. Since EZH2 KO primary tumors likely have to undergo similar processes to WT cells to metastasize, our results suggest that EZH2 deficiency in the former may not only negatively affect tumor cell mobilization and survival, but also the processes leading to metastasis. Prior studies have suggested multi-pronged roles of EZH2 in the modulation of TNBC aggressiveness [8,9,10,11].

TINs have emerged as key modulators of primary tumor growth and metastatic progression in TNBC [26]. As the most abundant leukocyte in human bone marrow and blood, neutrophils can infiltrate tumors in high numbers and acquire novel activities therein, promoting anti- and/or pro-tumorigenic functions [27]. In the context of TNBC patients, high levels of TINs are predictive of poor treatment responses and decreased survival [28,29]. TILs and TINs often play antagonistic roles, for example, TINs may inhibit the recruitment and/or activation of TILs via metabolic (e.g., arginase-mediated amino acid depletion) or cell–cell (e.g., PD-1/PD-L1 and TIM-3/Gal-9) interactions [30,31]. Because of these activities, TINs (whether derived from developmentally mature or immature neutrophils) are often categorized under the functional term “myeloid-derived suppressive cells” (MDSCs) [32].

Current therapeutic approaches face the challenge of alleviating cold tumor progression in the absence of T-cell activation. Although conventional anticancer treatments, like chemotherapy and radiotherapy, still have important roles to play in tumor burden reduction and preventing the selection of immune-resistant clones, the incorporation of improved therapies that suit the mutational burden in patients are much needed. In this context, increasing tumor sensitivity to ICI therapy by converting them from a “cold” to a “hot” phenotype may lead to better outcomes. Overall, limiting TIN infiltration to enable TIL antitumor activity (i.e., making cold tumors hot) is a major goal of current research [4]. Since EZH2 can be expressed in both TINs [33] and TILs [34], further understanding its role and impact as a tumor-intrinsic and/or immune-associated factor is critical to overcoming TNBC resistance and improving patient outcomes.

We acknowledge several limitations to the present study. First, our in vitro experiments included descriptive assessments of WT, EZH2 KO, and EZH2 OE 4T1 cells via proliferation, invasion, secretome, and surface flow cytometry assays, but did not include an extensive analysis of the epigenetic (e.g., by ATAC-Seq), transcriptomic (e.g., by RNA-Seq), proteomic, or metabolic (e.g., by mass spectrometry) processes of these cell lines. Based on the profound in vivo differences in the growth and metastasis of these cells observed in this study, future investigations are warranted in which WT, EZH2 KO, and EZH2 OE cells may be grown in vitro and sorted after in vivo expansion to compare their molecular makeup. Second, this study revealed significant effects of tumor-intrinsic EZH2 knockout on TIN/TIL poise, but did not provide functional and/or signaling data on these tumor-associated immune cells. Future investigations will tackle this issue after sorting individual subsets (neutrophils and CD8+ and CD4+ T cells, notably) using downstream analysis by RNA-Seq. Third, our findings with in vivo MSA-2 treatment suggest that the 4T1 WT, EZH2 KO, and OE cells are amenable to the combined testing of drugs directed at key pathways not only including STING, but also other EZH2-regulated transcriptional regulators, such as TGF β [14], STAT3 [15], and Wnt [16]. One may also envision investigating systemic treatment with candidate EZH2 inhibitors [17], although our study highlighted that tumor-intrinsic, rather than the global inhibition of this pathway, may be beneficial. Fourth, it is important to bear in mind that, while CRISPR editing as used in our study is efficient at targeting specific sites in the genome, it is also affected by potential off-target effects, which would need to be ascertained by deep sequencing methods in follow-up investigations. Fifth, our study assessed tumor growth in vivo every 3 days, but metastatic potential only at endpoint (21 or 28 days), and only in the lung. Future studies could measure metastatic potential at other timepoints and in other organs, such as the brain and liver. Ideally, it would be desirable in such extended studies to attempt a spatial transcriptomics analysis of human TNBC resection tissues from both primary tumor and metastatic sites to assess the potential association between EZH2 expression in tumor cells and the nearby presence of TILs vs. TINs, as suggested by our study using the 4T1 model.

## 4. Materials and Methods

### 4.1. Cell Lines

The parent WT 4T1 (CRL-2539) cell line [18] was purchased from ATCC and cultured at 37 °C and 5% CO_2_ in DMEM high-glucose medium (Sigma Aldrich, St Louis, MO, USA), supplemented with 10% fetal bovine serum (FBS, HyClone, purchased from Avantor, Radnor, PA, USA), 1% HEPES (Thermo Fisher Scientific, Hampton, NH, USA), 1% L-glutamine (Sigma Aldrich), and penicillin–streptomycin (100 U mL^−1^, Sigma Aldrich). Expression constructs were made using a modified GoldenGate Assembly protocol [35]. Murine EZH2 was amplified from pINTO-NFH:mEZH2 (Addgene #65925; gift from Roberto Bonasio [36]) and cloned into pBD170abc, a level 0 destination vector for the downstream cloning of an open reading frame. Then, a level 1 expression construct (pBD320) was cloned with a CMV promoter, BetaGlobin 3’UTR, into a position 1 destination vector (pTW324; Addgene #115955; gift from Ron Weiss) using Bsa1 and T4 DNA ligase. Puromycin alone (pBD324) or mScarlet-IRES[EMCV]-puromycin (pBD332) expression vectors under the control of a PGK promoter were cloned into a position 2 destination vector (pTW325; Addgene #115956; gift from Ron Weiss). Then, pBD320 was combined with a minimal linker and pBD324 or pBD332, respectively, to generate level 2 dual-expression constructs pBD321 and pBD333. For KO plasmids, the murine EZH2 sequences from AOI-WT-Cas9-sq-mouse Ezh2-E18-GFP and AOI-WT-Cas9-sq-mouse Ezh2-E10-GFP (Addgene #91880 and Addgene #91879; gift from Martine Roussel [37]) were cloned into pSPCas9(BB)-2A-Puro (Addgene #62988; gift from Feng Zhang [38]). Plasmids were transfected into 4T1 cells using lipofectamine 3000 (Thermo Fisher Scientific) following the manufacturer’s protocol and selected using puromycin at a concentration of 5 µg/mL. Stable OE clones were continually grown in puromycin; KO clones were only exposed to puromycin for up to a week. Cells were then isolated into 96-well plates and the clones selected. 

### 4.2. Western Blot

Cell lysates were prepared using the Minute Total Protein Extraction Kit (Invent Biotechnologies, Plymouth, MN, USA). The provided denaturing buffer was supplemented with Halt’s protease and phosphatase inhibitor cocktail at a 3× concentration (Thermo Fisher Scientific). Lysates were passed through spin columns to remove viscosity. Total protein concentration was measured using the Pierce Rapid Gold BCA protein assay kit (Thermo Fisher Scientific), concentration was normalized across treatments with denaturing buffer, then a Laemmli sample buffer (Bio-Rad Laboratories, Hercules, CA, USA) supplemented with β-mercaptoethanol was added. Samples were then boiled at 95 °C for 5 min and lysates were separated by SDS-PAGE on Any-KD gels (Bio-Rad Laboratories). Protein transfer to nitrocellulose membranes (LI-COR Biosciences, Lincoln, NE, USA) was performed at 4 °C by wet transfer in Towbin buffer (25 mM Tris, 192 mM glycine, and 20% MeOH (*v*/*v*) without SDS). After transfer, the membrane was rinsed three times with double-distilled H_2_O. Blocking was performed using Intercept (TBS) blocking buffer (LI-COR Biosciences) for one hour at room temperature in motion. Primary antibodies to EZH2 and α-tubulin (from Cell Signaling Technologies, Danvers, MA, USA, used at 1:1000) were added to an Intercept T20 antibody diluent (LI-COR) at 4 °C overnight in motion. Secondary IRDye 800CW donkey anti-rabbit IgG antibody (from LI-COR Biosciences, used at 1:15,000) was added to the Intercept antibody diluent (LI-COR Biosciences) for 1 h at 37 °C on a shaker. Membranes were analyzed using an Odyssey CLx imager and Image Studio software (version 5.5, LI-COR Biosciences).

### 4.3. Cell Proliferation Assay

EZH2 KO, EZH2 OE, and WT 4T1 cells were harvested, resuspended in DMEM-complete medium, and counted. Then, 5 × 10^5^ cells were plated on a T25 flask in 5 mL of medium and cultured at 37 °C and 5% CO_2_. At 24, 48, and 72 h, the cells were harvested and stained with propidium iodide to determine the cell count and viability using a hemocytometer.

### 4.4. 3D Spheroid Invasion Assay 

To generate EZH2 KO, EZH2 OE, and WT 4T1 cell spheroids, 3000 cells were plated in 200 µL on a Spheroid Nunclon 96-well plate (Thermo Scientific) and centrifuged at 450× *g* for 5 min at 4 °C and incubated at 37 °C and 5% CO_2_. After 48–72 h of incubation, spheroids were collected, embedded in 3 mg/mL collagen type I (Corning, Glendale, AZ, USA), and then plated in a 35 mm glass-bottom dish (Cellvis, Mountain View, CA, USA) for incubation overnight at 37 °C. After collagen was polymerized, complete DMEM (Thermo Fisher Scientific) was added to cover the collagen matrix and spheroids. An IX51 microscope (Olympus, Center Valley, PA, USA) at a 10× magnification (equal to 1.5 pixels/μm) with an Infinity2 charge-coupled device camera was used for 3D spheroid imaging. Spheroid circulatory and invasiveness were measured by ImageJ, https://imagej.net/ij/download.html (accessed on 1 August 2023), as previously described [19].

### 4.5. Extracellular Mediator Assay

Supernatants from in vitro cultures of EZH2 KO, EZH2 OE, and WT 4T1 cells were collected and stored at −80 °C until use. Extracellular mediators were quantified using a U-PLEX multiplexed chemiluminescent ELISA assay (Meso Scale Discovery, Rockville, MD, USA), following the manufacturer’s protocol. Plates were acquired on the QuickPlex SQ 120MM reader and later analyzed using Discovery Workbench 4.0 software (both from Meso Scale Discovery).

### 4.6. Animals

BALB/c mice (females, 6–8 weeks old) were purchased from Jackson Laboratories and maintained in the Division of Animal Resourcesfacilities at Emory University. Experiments were performed in accordance with the Emory University Institutional Animal Care and Use Committee’s approved protocol (DAR-2017-00-504). Female mice were chosen to match the strain and sex of origin of the parent WT 4T1 cell line. To establish the model, 5 × 10^5^ cells were injected subcutaneously (s.c.) in the right flank as the location of the primary tumor [39]. Tumor size (mm^2^) was measured in two dimensions with Vernier calipers every 3 days. In some cases, WT and EZH2 KO cells were injected s.c. and animals were concomitantly treated with the synthetic STING agonist MSA-2 [40]. MSA-2 (benzothiophene oxobutanoic acid; Cat: HY-136927) was purchased from MedChemExpress (Monmouth Junction, NJ, USA) and dissolved in 20% sulfobutylether-β-cyclodextrin (SBE-β-CD, Cat: HY-17031, MedChemExpress) in 0.9% saline to a concentration of 25 mg/mL. This stock was stored at −20 °C in the dark until injection. MSA-2 at 25 mg/kg in SBE-β-CD in a 200 µL volume was administered subcutaneously in the left flank. Doses were administered every third day [41], and the non-treated control group received 200 µL of SBE-β-CD only.

### 4.7. Lung Metastasis Assay

Lungs were isolated under sterile conditions from tumor-bearing mice 21–28 days post-injection, then minced and digested in 1 mg/mL of collagenase IV (Millipore Sigma, Burlington, MA, USA) for 2 h at 37 °C under a rotating motion. After digestion, single-cell suspensions were filtered through a 70 μm strainer and washed twice in selection medium consisting of complete DMEM with 6-thioguanine (Millipore Sigma) at 60 μM. Cells were resuspended in 8 mL of selection medium, and 1 mL was plated per well in a 6-well plate for each lung digestion. After 7 to 14 days of incubation in the selection medium (to kill lung fibroblasts without affecting tumor cells), as soon as one of the wells reached confluency, all wells were harvested and counted on a Cellometer T4 Automated Counter (Nexcelom, Lawrence, MA, USA) using trypan blue to discriminate dead cells.

### 4.8. Flow Cytometry Staining and Data Acquisition

For in vitro analyses, EZH2 KO, EZH2 OE, and WT 4T1 cells were thawed, resuspended in DMEM-complete medium, and cultured in T75 flasks. Before reaching confluency, the cells were harvested and counted using propidium iodide. For ex vivo analyses, tumors grown in the flank of mice were harvested, weighed, minced, and digested in liberase TL (Roche, Indianapolis, IN, USA) and DNAse (Roche) for 30 min at 37 °C in motion. Cell suspensions were filtered through a 70 μm strainer and washed with PBS. Total cell count was determined using a Cellometer T4 Automated Counter and trypan blue. All cells were pre-incubated with Fc receptor blocking antibody (Clone 24G2, BioLegend, San Diego, CA, USA) in an FACS buffer at room temperature for 10 min. Then, the cells were incubated with fluorochrome-conjugated antibodies for 30 min at 4 °C. The in vitro staining panel included antibodies to CD24 (clone M1/69), CD44 (clone IM7), CD80 (clone 16-10A1), ICAM-1 (clone YN1/1.7.4), MHC class I (clone M1/42), MHC class II (clone M5/114.15.2), and PD-L1 (clones 10F.9G2), all purchased from BioLegend. The ex vivo staining panel included, in addition to the above, antibodies to CD3 (clone 17A2), CD4 (clone GK15), CD8a (clone 53-6.7), CD11b (clone M1/70), CD11c (clone N418), CD19 (clone 6D5), CD45 (clone 30-F11), CD69 (clone H1.2F3), CD107a (clone 1D4B), F4/80 (clone BM8), Ly6C (clone HK1.4), Ly6G (clone 1A8), NK1.1 (clone PK136), and PD-1 (clone 29F.1A12), also purchased from BioLegend. The live dead fixable NIR (1:400 in PBS) was obtained from ThermoFisher Scientific. After incubation, the cells were washed three times with FACS buffer and analyzed using the Aurora Spectral Flow Cytometer (Cytek Biosciences, Fremont, CA, USA). Data were analyzed using FlowJo software (version 10.10, BD Biosciences, Franklin Lakes, NJ, USA).

### 4.9. Statistical Analysis

All statistical analysis and graphs were performed using Prism software (version 10.2, GraphPad Software, Boston, MA, USA). Non-parametric methods were used for descriptive statistics (box plots with median line and interquartile range forming outside boundaries to illustrate distributions), and comparisons between conditions and or timepoints. One-way ANOVA was used to analyze differences between the three groups (EZH2 KO, EZH2 OE, and WT 4T1 cells) at fixed timepoints. Two-way ANOVA was used to test group differences across timepoints, e.g., for invasiveness in 3D spheroid assay over 3 days in vitro, or primary tumor growth in the range of 21–28 days in vivo. Values of *p* < 0.05 were considered significant (* *p* < 0.05, ** *p* < 0.01, *** *p* < 0.001, and **** *p* < 0.0001).

## Figures and Tables

**Figure 1 ijms-25-05392-f001:**
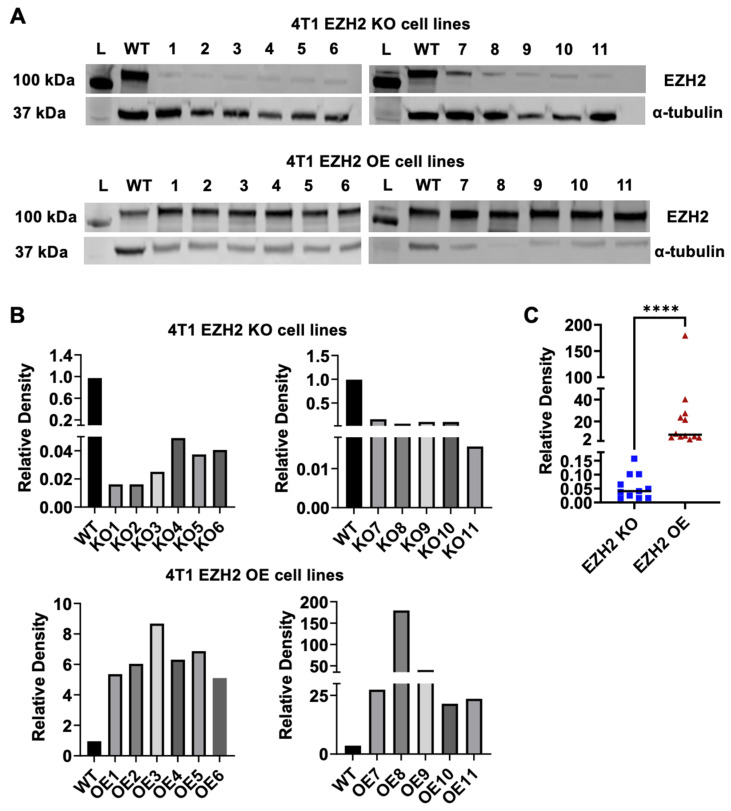
EZH2 knockout (KO) and overexpression (OE) cell clones were generated from the parent wild-type (WT) 4T1 TNBC line. (**A**) Western blots (L indicates protein ladder) and (**B**) densitometric analysis comparing 11 clones from each EZH2 KO (upper panel) and EZH2 OE (lower panel) lines to the 4T1 WT line (2 sets of blots for each), with alpha-tubulin as the normalization control. (**C**) Comparison of EZH2 protein expression between KO (blue squares) and OE (red triangles) groups by Wilcoxon rank-sum test, **** *p* < 0.0001.

**Figure 2 ijms-25-05392-f002:**
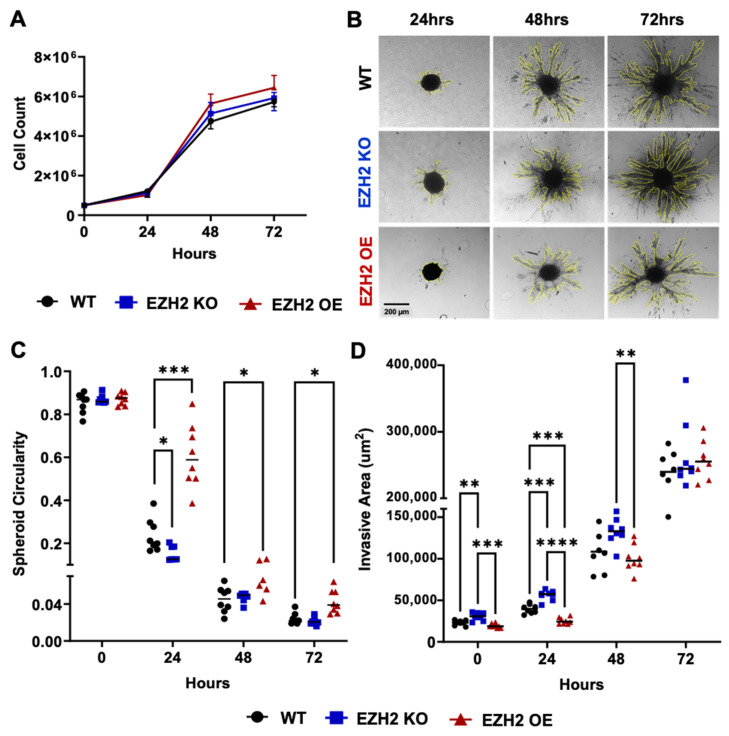
In vitro replicative and invasive behaviors of EZH2 KO and EZH2 OE compared to parent WT 4T1 cells. (**A**) Counts of 4T1 WT, EZH2 KO, and EZH2 OE lines over 72 h of growth in 2D plates. (**B**) Representative images of spheroid for 4T1 WT, EZH2 KO, and EZH2 OE lines, and quantification of (**C**) circularity and (**D**) invasive area over 72 h of growth in a 3D invasion assay (n = 8 spheroids per group). Comparisons across groups and timepoints are by two-way ANOVA with Tukey’s post-hoc test and shown as * *p* < 0.05, ** *p* < 0.01, *** *p* < 0.001, and **** *p* < 0.0001.

**Figure 3 ijms-25-05392-f003:**
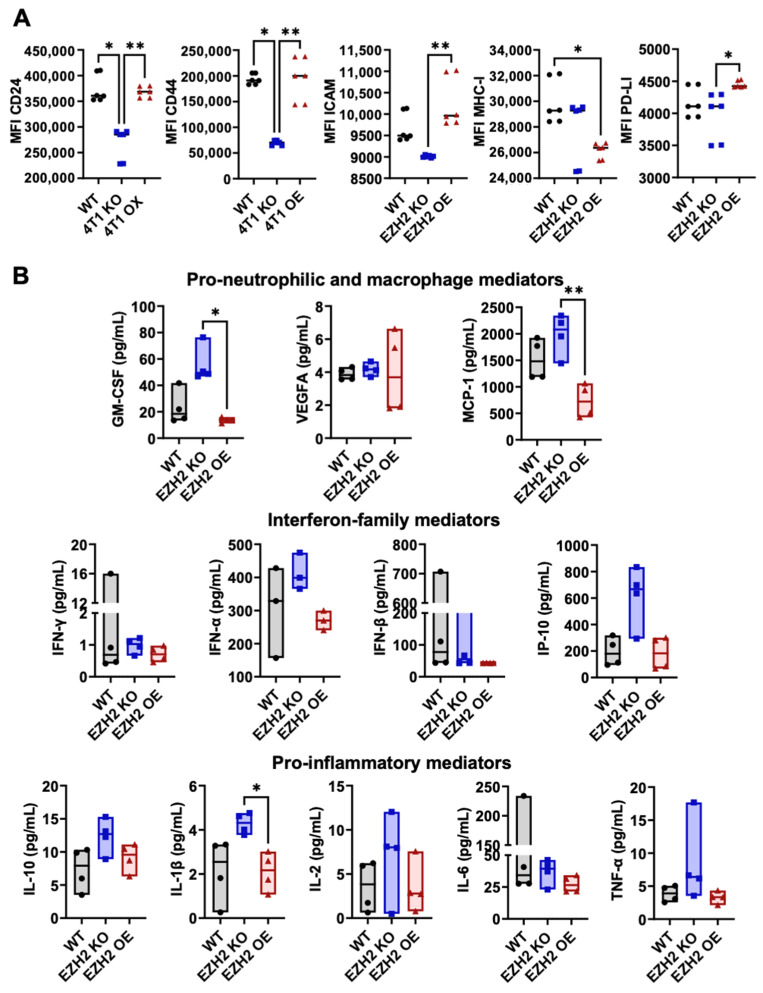
In vitro surface phenotype and secreted factors by EZH2 KO and EZH2 OE compared to parent WT 4T1 cells. (**A**) 4T1 WT (black circles), EZH2 KO (blue squares), and EZH2 OE (red triangles) lines were cultured in DMEM for 24 h and analyzed for the surface expression of relevant surface markers by flow cytometry (six repeats, see Methods and Appendix A for details). (**B**) Culture supernatants were screened for relevant extracellular mediators via mesoscale assay (four repeats). Comparison between groups is by one-way ANOVA with Tukey’s post-hoc test and shown as * *p* < 0.05, ** *p* < 0.01.

**Figure 4 ijms-25-05392-f004:**
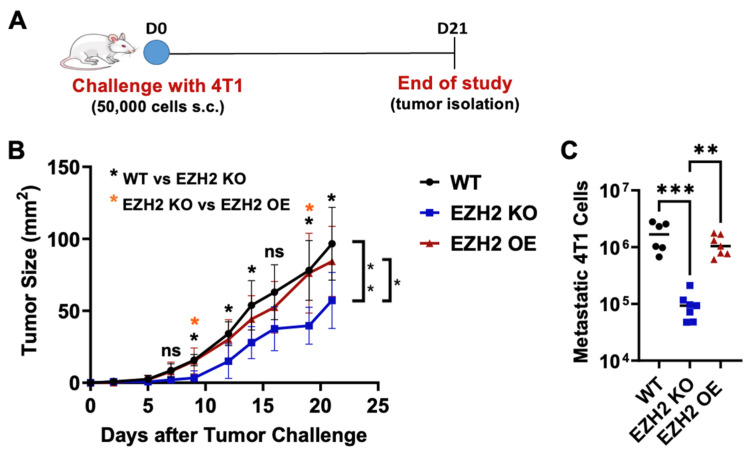
In vivo primary tumor growth and lung metastasis by EZH2 KO and EZH2 OE compared to parent WT 4T1 cells. (**A**) Experimental timeline of 4T1 WT, EZH2 KO, and EZH2 OE injections in mice. (**B**) Growth of 4T1 WT, EZH2 KO, and EZH2 OE primary tumors over 21 days post-injection (n = 6–7 mice per group). Comparisons across groups and timepoints are by two-way ANOVA with Tukey’s post-hoc test and shown as * *p* < 0.05 and ** *p* < 0.01 (brackets). Comparison between groups at each timepoint is by one-way ANOVA and shown as * *p* < 0.05 (as indicated for WT vs. EZH2 KO and EZH2 KO vs. EZH2 OE, above each timepoint). (**C**) Lung metastasis assays for 4T1 WT, EZH2 KO, and EZH2 OE lines. Comparisons between groups are by one-way ANOVA with Tukey’s post-hoc test and shown as ** *p* < 0.01 and *** *p* < 0.001.

**Figure 5 ijms-25-05392-f005:**
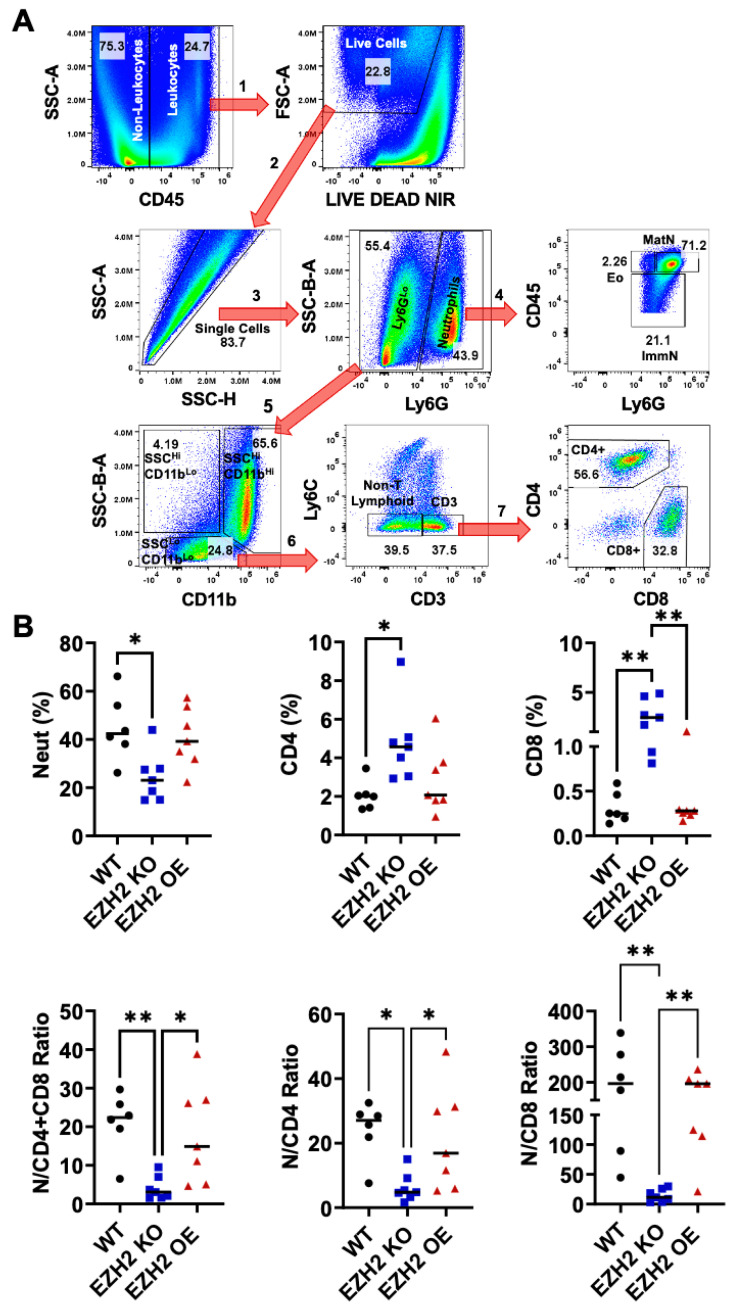
In vivo primary tumor infiltration by neutrophils and CD4+ and CD8+ T cells for EZH2 KO and EZH2 OE compared to parent WT 4T1 cells. (**A**) Flow cytometry strategy for gating of infiltrating leukocyte subsets in 4T1 WT, EZH2 KO, and EZH2 OE primary tumors, with sequential steps 1 (leukocytes), 2 (live cells), 3 (singlets), 4 (granulocytes, including mature neutrophils, immature neutrophils, and eosinophils), 5 (non-granulocytes), 6 (T cells), and 7 (CD4+ and CD8+). (**B**) Relative percentages of infiltrating neutrophils and CD4+ and CD8+ cells among live leukocytes (top), and ratios between these subsets (bottom) in 4T1 WT (black circles), EZH2 KO (blue squares), and EZH2 OE (red triangles) primary tumors. Comparisons between groups are by one-way ANOVA with Tukey’s post-hoc test and shown as * *p* < 0.05 and ** *p* < 0.01.

## Data Availability

Data are available upon reasonable request from the authors.

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
