# Peer review of "Tumor-Intrinsic Enhancer of Zeste Homolog 2 Controls Immune Cell Infiltration, Tumor Growth, and Lung Metastasis in a Triple-Negative Breast Cancer Model"

_ijms, 2024, doi:10.3390/ijms25105392_

Round 1

Reviewer 1 Report

Comments and Suggestions for Authors

The manuscript by Lenore Monterroza et al. “Tumor-Intrinsic Enhancer of Zeste Homolog 2 Controls Immune Cell Infiltration, Tumor Growth and Lung Metastasis in a Triple Negative Breast Cancer Model” demonstrated that a tumor-intrinsic role of EZH2 in the 4T1 TNBC model in regulating TIN/TIL poise, primary tumor progression and lung metastasis in vivo. Although authors have done many experiments to support their findings, there are some limitations as mentioned below: 

1. The authors should show the tumor figures in the manuscript.

2. The authors should do more cellular and molecular experiments to detect replicative and invasive behaviors.

3. The authors should improve the quality of the bands of Western Blot.

4. The grammar and word errors need to be corrected thoroughly in the manuscript.

Comments on the Quality of English Language

The grammar and word errors need to be corrected thoroughly in the manuscript.

Reviewer 2 Report

Comments and Suggestions for Authors

The study examines Triple-Negative Breast Cancer (TNBC) with an emphasis on EZH2’s function, a methyltransferase enzyme, in the proliferation of tumor cells and the penetration of immune cells. The study is interesting but there are some issues to address.

1. Figure 1b, asterisk marks for statistical significance are missing.

2. In Figure 2b, spheroid formation decreases in the OE group. This observation prompts the question of whether this could be interpreted as an anticancer effect. Additionally, despite assertions in the background, EZH2 does not seem to be associated with metastatic TNBC. Therefore, a scientific discourse should be incorporated to explain why this phenomenon is not observed.

3. Figure 3, most mediators in the EZH2 OE group are similar to the expression of wild type. Since it is clear that the wild type cells themselves are malignant, it can be interpreted that expressing the EZH2 will not have a relationship with aggressive TNBC.

4. The in vivo anticancer results in Figure 4 also have illogical point in the same context as Figure 3. Further, a macroscopic analysis data, such as IHCs or photos, for representative of the results in Figure 4C must be provided.

Comments on the Quality of English Language

Minor langulage polishing is required.

Reviewer 3 Report

Comments and Suggestions for Authors

This study investigated the role of enhancer of EZH2, a pro-tumor methyltransferase, in TNBC using the 4T1 murine TNBC model. The authors generated EZH2 KO and oOE cell lines from the parent wild-type (WT) 4T1 cells using CRISPR technology. In vitro experiments showed little change in replicative capacity and invasiveness of EZH2 KO and OE cells compared to WT cells, but marked changes in surface marker profile and cytokine/chemokine secretion were observed.

In vivo experiments revealed that EZH2 KO cells showed significantly reduced primary tumor growth and a 10-fold decrease in lung metastasis compared to WT cells, while EZH2 OE cells were unchanged. The tumor-infiltrating neutrophil to tumor-infiltrating lymphocyte (TIN:TIL) ratios were greatly reduced in EZH2 KO tumors but remained unchanged in EZH2 OE tumors compared to WT tumors. These findings demonstrate a tumor-intrinsic role of EZH2 in the 4T1 TNBC model in regulating TIN/TIL balance, primary tumor progression, and lung metastasis in vivo.

  1. While the study demonstrates the effects of EZH2 KO and OE on tumor growth and metastasis, it does not provide clear mechanistic insights into how EZH2 regulates these processes.
  2. The authors should perform more comprehensive analyses of the generated cell lines, such as transcriptomic and proteomic profiling, to better understand the molecular changes induced by EZH2 manipulation.
  3. The study only assessed tumor growth and metastasis at one timepoint (21 or 28 days post-injection).
  4. To translate the findings into potential therapeutic strategies, the authors should consider using EZH2 inhibitors in their experiments and compare the results with EZH2 KO.
  5. While the study quantified TINs and TILs in the tumors, it did not assess their functional states.
  6. Incorporating in vivo imaging techniques, such as bioluminescence or fluorescence imaging, would allow for real-time monitoring of tumor growth and metastasis, providing more dynamic and sensitive measurements.
  7. To strengthen the clinical relevance of the findings, the authors should consider validating their results using human TNBC samples, such as examining EZH2 expression levels and correlating them with TIN/TIL ratios and patient outcomes.
  8. The study does not adequately address the potential off-target effects of CRISPR-mediated EZH2 KO and OE.

Round 2

Reviewer 2 Report

Comments and Suggestions for Authors

All concerns have been well addressed. There is no additional issue to raise.

Comments on the Quality of English Language

Minor language polishing is required.